# Second-Order Adversarial Attack and Certifiable Robustness

## Abstract

Adversarial training has been recognized as a strong defense against adversarial attacks. In this paper, we propose a powerful second-order attack method that reduces the accuracy of the defense model by Madry et al. (2017). We demonstrate that adversarial training overfits to the choice of the norm in the sense that it is only robust to the attack used for adversarial training, thus suggesting it has not achieved universal robustness. The effectiveness of our attack method motivates an investigation of provable robustness of a defense model. To this end, we introduce a framework that allows one to obtain a certifiable lower bound on the prediction accuracy against adversarial examples. We conduct experiments to show the effectiveness of our attack method. At the same time, our defense model achieves significant improvements compared to previous works under our proposed attack.

## 1 Introduction

Deep neural networks (DNNs) have achieved significant success when applied to a variety of challenging machine learning tasks. For example, DNNs have obtained state-of-the-art accuracy on large-scale image classification (Krizhevsky et al., 2012; He et al., 2016b). At the same time, vulnerability to adversarial examples, an undesired property of DNNs, has drawn attention in the deep-learning community (Szegedy et al., 2013; Goodfellow et al., 2014). Generally speaking, adversarial examples are perturbed versions of the original data that successfully fool a classifier. For example, in the image domain, adversarial examples are images transformed from natural images with visually negligible changes but lead to different classification results (Goodfellow et al., 2014). The existence of adversarial examples has raised many concerns, especially in scenarios with a high risk of misclassification, such as autonomous driving.

To tackle adversarial examples, many works have been proposed to improve the robustness of DNNs (Papernot et al., 2016; Meng & Chen, 2017), referred to as defense models. However, most of these defense methods have been later attacked successfully by new attack methods (Carlini & Wagner, 2017b;a). For example, Athalye et al. (2018) conducted a case study that successfully attacked seven defense methods submitted to ICLR2018.

One model that demonstrated good performance against strong attacks, and has thus far not been successfully attacked, is based on adversarial training (Goodfellow et al., 2014; Madry et al., 2017). Adversarial training constructs a defense model by augmenting the training set with adversarial examples. Though successful in adversarial defensing, the underlying mechanism is still unclear. In this paper, we explain the good performance is due to "degenerate global minimum", a phenomenon firstly discussed in (Tramèr et al., 2017), and show that the Madry's defense model (Madry et al., 2017), adversarial training with $\ell_\infty$ attack is not robust against an $\ell_2$ attack even on MNIST. In particular, we develop a new attack method based on the approximated second-order derivative that forces its accuracy worse than naturally trained baseline models. To our knowledge, this is the first $\ell_2$ attack method that significantly reduces the accuracy of Madry's model. In addition, the $\ell_\infty$ version of our method also breaks the $\ell_2$ based adversarially trained model. Considering that Sharma & Chen (2017) has proposed $\ell_1$-based adversarial examples that break the Madry's model, we believe the robustness of adversarial training overfits to the choice of norms and is not universally robust. Concurrent work by Schott et al. (2018) also proposes similar conclusions via comprehensive experiments.

Our findings lead to a concern that most of the existing defense methods are heuristically driven, thus any defense method that cannot provide theoretically-provable robustness guarantee is potentially vulnerable to future attacks.

Several works on provable or certifiable adversarial defense methods have been proposed recently (Raghunathan et al., 2018; Kolter & Wong, 2017; Sinha et al., 2017). However, most of them either have to make strong assumptions, such as on the structure of the model or on the smoothness of the loss function or are difficult to extend to large-scale datasets, the most common application scenario for DNNs.

Recently, Mathias et al. (2018) developed theoretical insight of certifiable robust prediction, by building a connection between differential privacy and model robustness. It is shown that adding properly chosen noise to the classifier will lead to certifiable robust prediction.

Built on the idea in (Mathias et al., 2018), as our second contribution, we conduct an analysis based on Rényi divergence (Van Erven & Harremos, 2014) and show a higher upper bound on the tolerable size of attacks compared with Mathias et al. (2018). In addition, we suggest there exists a connection between adversarial defense and robustness to random noise. Based on this, we introduce a more comprehensive framework, incorporating stability training to improve classification accuracy. One advantage of our framework is that it has no requirement on the model structure and is applicable to all classification models. Considering MNIST and CIFAR-10, our experiments demonstrate that the proposed defense yields stronger robustness to adversarial attacks, compared to other models.

## 2 PRELIMINARY

### 2.1 NOTATION

We consider the task of image classification. Natural images are represented by $\mathbf{x} \in \mathcal{X} \triangleq [0, 1]^{h \times w \times c}$, where $\mathcal{X}$ represents the image space, with $h, w, c$ the height, width, and channels of an image, respectively. An image classifier over $k$ classes is considered as a function $f : \mathcal{X} \to \{1, \ldots, k\}$. In this paper, we only consider classifiers constructed by DNNs. To better present our framework, we define a stochastic classifier, a function $f$ over $\mathbf{x}$ with output $f(\mathbf{x})$ being a multinomial distribution over $\{1, \ldots, k\}$, i.e., $P(f(\mathbf{x}) = i) = p_i$ for $\sum_i p_i = 1$. One can classify $\mathbf{x}$ by picking $\arg\max_i p_i$. Note this distribution is different from the one generated from softmax.

### 2.2 RÉNYI DIVERGENCE

Our theoretical result depends on the Rényi divergence, defined as follows (Van Erven & Harremos, 2014):

**Definition 1 (Rényi Divergence)** *For two probability distributions $P$ and $Q$ over $\mathcal{R}$, the Rényi divergence of order $\alpha > 1$ is*

$$D_\alpha(P\|Q) = \frac{1}{\alpha - 1} \log \mathbb{E}_{x \sim Q} \left( \frac{P}{Q} \right)^\alpha \tag{1}$$

### 2.3 ADVERSARIAL EXAMPLES

Given a classifier $f : \mathcal{X} \to \{1, \ldots, k\}$ for an image $\mathbf{x} \in \mathcal{X}$, an adversarial example $\mathbf{x}'$ satisfies $\mathcal{D}(\mathbf{x}, \mathbf{x}') < \epsilon$ for some small $\epsilon > 0$, and $f(\mathbf{x}) \neq f(\mathbf{x}')$, where $\mathcal{D}(\cdot, \cdot)$ is some distance metric, i.e., $\mathbf{x}'$ is close to $\mathbf{x}$ but yields a different classification result. The distance is often described in terms of an $\ell_p$ metric, and in most of the literature $\ell_2$ and $\ell_\infty$ metrics are considered. In this paper, we focus on the $\ell_2$ metric, but our method is easy to extend to the $\ell_\infty$ scenario.

Adversarial examples are often constructed by iterative optimization methods. Previous work has proposed a number of adversarial attack methods, such as the Fast Gradient Sign Method (FGSM) (Kurakin et al., 2016), along with its multi-step variant FGSM$^k$, which is equivalent to exploring adversarial examples that increase the classification loss using projected gradient descent (PGD) (Madry et al., 2017):

$$\mathbf{x}^{t+1} = \Pi_{\mathbf{x}+\mathcal{S}}(\mathbf{x}^t + \alpha(\nabla_{\mathbf{x}} L(\theta, \mathbf{x}, y))) \tag{2}$$

where $\Pi_{\mathbf{x}+\mathcal{S}}$ is the projection operation that ensures adversarial examples stay in the $\ell_p$ ball $\mathcal{S}$ around $\mathbf{x}$. In (Madry et al., 2017), it has also been shown that a PGD attack is a universal adversary

among all first-order attack methods. Their analysis and evidence from experiments all suggest any adversarial attack method that only incorporates gradients of the loss function w.r.t. the input cannot do significantly better than PGD.

## 2.4 ADVERSARIAL TRAINING

Adversarial training constructs adversarial examples and includes them into a training set to train a new and more robust classifier. This method is intuitive and has gained great success in defense (Goodfellow et al., 2014; Madry et al., 2017). The motivation behind adversarial training is that finding a robust model against adversarial examples is equivalent to solving the saddle-point problem $\min_\theta \max_{\mathbf{x}':D(\mathbf{x},\mathbf{x}')<\epsilon} L(\theta, \mathbf{x}', y)$. The inner maximization is equivalent to constructing adversarial examples, while the outer minimization is the standard training procedure for loss minimization.

## 3 SECOND-ORDER ADVERSARIAL ATTACK

In this section, we propose an efficient second-order adversarial attack method. As one motivation, note most current attack methods construct adversarial examples based on the gradient of a loss function. However, a first-order derivative is not effective for attacks if the defense model is trained adversarially.

To see this, first note that adversarial training is equivalent to solving the optimization problem: $(\hat{\theta}, \hat{\mathbf{x}}) = \operatorname{argmin}_\theta \operatorname{argmax}_{\mathbf{x}':D(\mathbf{x},\mathbf{x}')<\epsilon} L(\theta, \mathbf{x}', y)$. The solution is a saddle point, *i.e.*, it not only converges to a local minimum $\hat{\theta}$ in the parameter space, but also converges to a local maximum $\hat{\mathbf{x}}$ in the sample space, in which the gradient ideally vanishes at $\hat{\mathbf{x}}$ as $\nabla_{\mathbf{x}} L(\hat{\theta}, \mathbf{x}, y)|_{\hat{\mathbf{x}}} = 0$. In practice, an adversarial training often finds $\hat{\theta}$ that makes the loss function flat in the neighborhood of a natural example $\mathbf{x}$, which leads to inefficient exploration for adversarial examples in the attack methods. This phenomenon is referred to as "degenerate global minimum" in (Tramèr et al., 2017) and analyzed for adversarial training via single-step attacks. Here we argue using multiple-step attacks also suffers from this issue. This motivates utilization of the second-order derivative of the loss function to construct adversarial examples.

Specifically, assume the loss function is twice differentiable with respect to $\mathbf{x}$. Using Taylor expansion on the difference between the losses on the original and perturbed samples, and assuming the gradient vanishes, we have $L(\theta, \mathbf{x} + \mathbf{r}, y) - L(\theta, \mathbf{x}, y) \approx \frac{1}{2}\mathbf{r}^T H(\theta, \mathbf{x}, y)\mathbf{r}$ with $\mathbf{r}$ being the perturbation, and $H(\theta, \mathbf{x}, y)$ is the Hessian matrix of the loss function. Our goal is to find a small perturbation $\mathbf{r}$ that maximizes the difference $L(\theta, \mathbf{x} + \mathbf{r}, y) - L(\theta, \mathbf{x}, y)$. Our idea is based on the observation that the optimal perturbation direction should be in the same direction as the first dominant eigenvector, $\mathbf{e}(\theta, \mathbf{x}, y)$, of $H(\theta, \mathbf{x}, y)$, that is $\mathbf{r} = \epsilon \frac{\mathbf{e}(\theta,\mathbf{x},y)}{\|\mathbf{e}(\theta,\mathbf{x},y)\|_2}$ for some constant $\epsilon > 0$. However, computing the eigenvectors of the Hessian matrix requires $O(I^3)$ runtime with $I$ the dimension of the data. To tackle this issue, we adopt the fast approximation method from Miyato et al. (2017), which is essentially a combination of the power-iteration method and the finite-difference method, to efficiently find the direction of the eigenvector. Based on this method, the optimal direction, denoted $\mathbf{r}_{adv}$, is approximated* by

$$\mathbf{r}_{adv} = \frac{g}{\|g\|_2}, \quad \text{with } g = \nabla_{\mathbf{x}} L(\theta, \mathbf{x}, y)|_{\mathbf{x}+\xi\mathbf{d}} \tag{3}$$

where $\mathbf{d}$ is a randomly sampled unit vector and $\xi > 0$ is a manually chosen step size. In practice, $\mathbf{d}$ is drawn from a centered Gaussian distribution and normalized such that its $\ell_2$ norm is 1.

This procedure is essentially a stochastic approximation to the optimal direction, where the randomness comes from $\mathbf{d}$. To reduce the variance of the approximation, we further take the expectation over the Gaussian noise, yielding $g = \mathbb{E}_{\mathbf{d}\sim N(0,\sigma^2 I)} [\nabla_{\mathbf{x}} L(\theta, \mathbf{x}, y)|_{\mathbf{x}+\mathbf{d}}]$. Note that choosing $\sigma$ is equivalent to choosing the step size $\xi$ in (3). Finally, we construct adversarial examples by an iterative update via PGD:

$$\mathbf{x}^{t+1} = \Pi_{\mathbf{x}+\mathcal{S}}(\mathbf{x}^t + \alpha\mathbf{r}_{adv}) = \Pi_{\mathbf{x}+\mathcal{S}}(\mathbf{x}^t + \alpha\frac{g^t}{\|g^t\|_2}), \quad \text{where } g^t = \mathbb{E}_{\mathbf{d}\sim N(0,\sigma^2 I)} [\nabla_{\mathbf{x}} L(\theta, \mathbf{x}, y)|_{\mathbf{x}+\mathbf{d}}] \tag{4}$$

---

*Detailed derivations are provided in the Appendix.

Intuitively, this method perturbs the example at each iteration and tries to move out of the local maximum in the sample space, due to the introduction of random Gaussian noise.

**Connection to Expectation of Transformation (EOT) Attack**   We can think of (4) as gradient descent for maximizing an objective function of $\mathbb{E}_{\mathbf{d}\sim N(0,\sigma^2 I)}\left[\nabla_\mathbf{x} L(\theta, \mathbf{x}, y)|_{\mathbf{x}+\mathbf{d}}\right]$. Note $\mathbb{E}_{\mathbf{d}\sim N(0,\sigma^2 I)}\left[\nabla_\mathbf{x} L(\theta, \mathbf{x}, y)|_{\mathbf{x}+\mathbf{d}}\right] = \nabla_\mathbf{x}\mathbb{E}_{\mathbf{d}\sim N(0,\sigma^2 I)}\left[\nabla_\mathbf{x} L(\theta, \mathbf{x}, y)|_{\mathbf{x}+\mathbf{d}}\right]$. Consequently, (4) is equivalent to constructing adversarial examples with the EOT method proposed by Athalye & Sutskever (2017), when a defense model contains added Gaussian noise $\mathbf{d}$.

In general, EOT tries to construct adversarial samples against a randomized defense model by solving the optimization problem: $\mathbf{x}' = \operatorname{argmax}_\mathbf{x} \mathbb{E}_{t\sim T}\left[L(\theta, \mathbf{x}, y)\right]$, where $t \sim T$ is the randomization in the defense model. Intuitively, EOT aims to estimate the correct gradient by excluding the effect of randomization via expectation. If a defense model tries to add Gaussian noise to its inputs, the corresponding optimization problem for EOT becomes $\mathbf{x}' = \operatorname{argmax}_\mathbf{x} \mathbb{E}_{\mathbf{d}\sim N(\mathbf{0},\sigma^2 I)}\left[\nabla_\mathbf{x} L(\theta, \mathbf{x}, y)|_{\mathbf{x}+\mathbf{d}}\right]$, which is the same as the approximate solution to our second-order attack method.

Note a significant difference between our method and EOT is that the random noise added in EOT depends on its corresponding defense model, whereas randomness in our method is independent of a defense model. In the experiments, we show our attack method dramatically reduces the accuracy of a defense model proposed by Madry et al. (2017).

## 4   CERTIFIABLE ROBUSTNESS

The effectiveness of the proposed attack method, as we will show in the experiments, suggests defending from adversarial attack is extremely difficult, and any defense model without *certifiable* robustness are always possible to be bypassed by stronger attacks.

Inspired by the PixelDP method (Mathias et al., 2018), we propose a framework that enables certifiable robustness on any classifier. Intuitively, our approach adds random noise to pixels of adversarial examples before classification, to eliminate the effects of adversarial perturbations. The most important feature of this framework is that it allows calculation of an upper bound on the tolerable size of attacks.

---

**Algorithm 1** Certifiable Robust Classifier

---

**Require:** An input image $\mathbf{x}$; A standard deviation $\sigma > 0$; A classifier $f$ over $\{1, \ldots, k\}$; Number of iterations $n$ ($n = 1$ is sufficient if only the robust classification $c$ is desired).
1: Set $i = 1$.
2: **for** $i \in [n]$ **do**
3:    Add i.i.d. Gaussian noise $N(0, \sigma^2)$ to each pixel of $\mathbf{x}$ and apply the classifier $f$ on it. Let the output be $c_i = f(\mathbf{x} + N(\mathbf{0}, \sigma^2 I))$.
4: **end for**
5: Estimate the distribution of the output as $p_j = \frac{\#\{c_i=j:i=1,\ldots,n\}}{n}$.
6: Calculate the upper bound

$$L = \sqrt{\sup_{\alpha>1}\left(-\frac{2\sigma^2}{\alpha} - \log\left(1 - p_{(1)} - p_{(2)} + 2\left(\frac{1}{2}\left(p_{(1)}^{1-\alpha} + p_{(2)}^{1-\alpha}\right)\right)^{\frac{1}{1-\alpha}}\right)\right)}$$

   where $p_{(1)}$ and $p_{(2)}$ are the first and the second largest values in $p_1, \ldots, p_k$.
7: Return classification result $c = \operatorname{argmax}_i p_i$ and the tolerable size of the attack $L$.

---

Our approach is summarized in Algorithm 1. In the following, we develop theory to prove the certifiable robustness of the proposed algorithm. Our goal is to show that if the classification of $\mathbf{x}$ in Algorithm 1 is $c$, then for any examples $\mathbf{x}'$ such that $\|\mathbf{x} - \mathbf{x}'\|_2 \leq L$, the classification of $\mathbf{x}'$ is also $c$.

To prove our claim, first recall that a stochastic classifier $f$ over $\{1, \ldots, k\}$ is a classifier whose output $f(\mathbf{x})$ has a multinomial distribution over $\{1, \ldots, k\}$ with probabilities as $(p_1, \ldots, p_k)$. In this context, robustness to an adversarial example $\mathbf{x}'$ generated from $\mathbf{x}$ means $\operatorname{argmax}_i p_i = \operatorname{argmax}_j p'_j$ with $P(f(\mathbf{x}) = i) = p_i$ and $P(f(\mathbf{x}') = j) = p'_j$, where $P(\cdot)$ denotes the probability of a specific output value. In the remainder of this section, we show Algorithm 1 achieves such robustness based on the Rényi divergence, starting with the following lemma.

**Lemma 1** *Let $P = (p_1, \ldots, p_k)$ and $Q = (q_1, \ldots, q_k)$ be two multinomial distributions over the same index set $\{1, \ldots, k\}$. If the indexes of the largest probabilities do not match on $P$ and $Q$, that is $argmax_i \, p_i \neq argmax_j \, q_j$, then*

$$D_\alpha(Q\|P) \geq -\log\left(1 - p_{(1)} - p_{(2)} + 2\left(\frac{1}{2}\left(p_{(1)}^{1-\alpha} + p_{(2)}^{1-\alpha}\right)\right)^{\frac{1}{1-\alpha}}\right) \quad (5)$$

*where $p_{(1)}$ and $p_{(2)}$ are the largest and the second largest probabilities in $p_i$'s.*

To simplify notation, we define $M_p(x_1, \ldots, x_n) = \left(\frac{1}{n}\sum_{i=1}^n x_i^p\right)^{1/p}$ as the generalized mean. The RHS in condition (5) becomes $-\log\left(1 - 2M_1\left(p_{(1)}, p_{(2)}\right) + 2M_{1-\alpha}\left(p_{(1)}, p_{(2)}\right)\right)$.

Lemma 1 proposes a lower bound of the Rényi divergence for changing the index of the maximum of $P$, *i.e.*, for any distribution $Q$, if $D_\alpha(Q\|P) < -\log\left(1 - 2M_1\left(p_{(1)}, p_{(2)}\right) + 2M_{1-\alpha}\left(p_{(1)}, p_{(2)}\right)\right)$, the index of maximum of $P$ and $Q$ must be the same. Based on Lemma 1, we obtain our main theorem on certifiable robustness, validating our claim:

**Theorem 2** *Suppose we have $\mathbf{x} \in \mathcal{X}$, and a potential adversarial example $\mathbf{x}' \in \mathcal{X}$ such that $\|\mathbf{x} - \mathbf{x}'\|_2 \leq L$. Given a $k$-classifier $f : \mathcal{X} \to \{1, \ldots, k\}$, let $f(\mathbf{x} + N(\mathbf{0}, \sigma^2 I)) \sim (p_1, \ldots, p_k)$ and $f(\mathbf{x}' + N(\mathbf{0}, \sigma^2 I)) \sim (p_1', \ldots, p_k')$.*

*If the following condition is satisfied, with $p_{(1)}$ and $p_{(2)}$ being the first and second largest probabilities in $p_i$'s:*

$$\sup_{\alpha > 1}\left(-\frac{2\sigma^2}{\alpha}\log\left(1 - 2M_1\left(p_{(1)}, p_{(2)}\right) + 2M_{1-\alpha}\left(p_{(1)}, p_{(2)}\right)\right)\right) \geq L^2, \quad (6)$$

*then $argmax_i \, p_i = argmax_j \, p_j'$*

With Theorem 2, we can enable certifiable robustness on any classifier $f$ by adding i.i.d. Gaussian noise to pixels of inputs during testing, as done in Algorithm 1. It provides an upper bound for the tolerable size of attacks for a classifier. Since the upper bound only depends on the output distribution $(p_1, \ldots, p_k)$, one can evaluate the bound based only on natural examples. Note the evaluation requires adjustment and computing confidence intervals for $p_{(1)}$ and $p_{(2)}$, but we omit the details as it is a standard statistical procedure. In the experiments, we use the end points of the $95\%$ confidence intervals for $p_{(1)}$ and $p_{(2)}$.

Based on the property of *generalized mean*, one can show that the upper bound is larger when the difference between $p_{(1)}$ and $p_{(2)}$ becomes larger. This is consistent with the intuition that a larger difference between $p_{(1)}$ and $p_{(2)}$ indicates more confident classification. In other words, more confident classification, in the sense that more probability concentrated on one class, is beneficial for robustness.

It is worth mentioning that, as pointed out in (Mathias et al., 2018), the noise is not necessarily added directly to the inputs but also to the first layer of a DNN. Given the Lipschitz constant of the first layer, one can still calculate an upper bound using our analysis. We omit the details here for simplicity.

Compared to the upper bound in PixelDP, our upper bound is strictly higher. We show the improvement using a simulation detailed in the Appendix. In the next section, we propose a simple strategy to improve the empirical performance of this framework.

## 5 Improved Certifiable Robustness

In Algorithm 1, for a classifier $f$ with a standard DNN, the added Gaussian noise is harmful to the classification accuracy on the original data. As discussed above, inaccurate prediction, in the sense that $p_{(1)}$ and $p_{(2)}$ are close, leads to weak robustness. Fortunately, one strength of Algorithm 1 is it requires nothing particular on the classifier $f$, which yields flexibility to modify $f$ to make it more robust against Gaussian noise.

Note robustness to Gaussian noise is much easier to achieve than robustness to carefully crafted adversarial examples. In PixelDP, the authors incorporated noise by directly adding the same noise during the training procedure. However, we note that there have been notable efforts at developing

robust DNNs to natural perturbations (Xie et al., 2012; Zhang et al., 2017); yet, these methods failed to defend models from adversarial attacks, as they are not particularly designed for that task. Our framework allows us to adapt these methods to improve the accuracy of classification when Gaussian noise is present, improving the robustness of our model. We emphasize that a connection between robustness to adversarial examples and robustness to natural perturbation (Xie et al., 2012; Zhang et al., 2017) has been established, which introduces a much wider scope of literature into the adversarial defense community.

**Stability Training**   The idea of introducing perturbations during training to improve model robustness has been studied in many works. In (Bachman et al., 2014) the authors regard perturbing models as a construction of pseudo-ensembles, to improve semi-supervised learning. More recently, Zheng et al. (2016) used a similar training strategy, named stability training, to improve classification robustness on noisy images.

For any natural image $\mathbf{x}$, stability training encourages its perturbed version $\mathbf{x}'$ to yield a similar classification result under a classifier $f$, *i.e.*, $D(f(\mathbf{x}), f(\mathbf{x}'))$ is small for some distance measure $D$. Specifically, given a loss function $L_0$ for the original classification task, stability training introduces a regularization term $L(\mathbf{x}, \mathbf{x}') = L_0 + \gamma L_{\text{stability}}(\mathbf{x}, \mathbf{x}') = L_0 + \gamma D(f(\mathbf{x}), f(\mathbf{x}'))$, where $\gamma$ controls the strength of the stability term. As we are interested in a classification task, we use cross-entropy as the distance $D$ between $f(\mathbf{x})$ and $f(\mathbf{x}')$, yielding the stability loss $L_{\text{stability}} = -\sum_j P(y_j|\mathbf{x}) \log P(y_j|\mathbf{x}')$ where $P(y_j|\mathbf{x})$ and $P(y_j|\mathbf{x}')$ are the probabilities generated after softmax. In this paper, we add i.i.d. Gaussian noise to each pixel of $\mathbf{x}$ to construct $\mathbf{x}'$, as suggested in (Zheng et al., 2016). Note that this is in the same spirit as adversarial training, but is only designed to improve the classification accuracy under a Gaussian perturbation.

## 6   EXPERIMENTS

We conduct experiments to evaluate the performance of our proposed methods in terms of attack effectiveness and defense robustness. Our methods are tested on the MNIST and CIFAR-10 data sets. The architecture of our model follows the ones used in (Madry et al., 2017). Specifically, for the MNIST data set, the model contains two convolutional layers with $32$ and $64$ filters, each followed by $2 \times 2$ max-pooling, and a fully connected layer of size $1024$. For the CIFAR-10 dataset, we use a wide ResNet model (Zagoruyko & Komodakis, 2016; He et al., 2016a). The network contains $5$ residual unites with $(16, 160, 320, 640)$ filters. The implementation details are provided in the Appendix. In both datasets, image intensities are scaled to $[0, 1]$, and the size of attacks are also rescaled accordingly. For reference, a distortion of $0.031$ in the $[0, 1]$ scale corresponds to $8$ in $[0, 255]$ scale.

### 6.1   THEORETICAL BOUND

We first evaluate the theoretical bounds of the size of attacks. With Algorithm 1, we are able to classify a natural image $\mathbf{x}$ and calculate an upper bound for the size of attacks $L$ for this particular image. Thus, with a given size of the attack $L_0$, we know the classification result is robust if $L_0 < L$.

Further, if the classification for a natural example is correct and robust for $L_0$ simultaneously, we know any adversarial examples $\mathbf{x}'$ such that $\|\mathbf{x} - \mathbf{x}'\|_2 < L_0$ will be classified correctly. Therefore, we can calculate the proportion of such examples in the test set to determine a lower bound of accuracy under attack of size $L_0$. We plot different lower bounds for various $L_0$ for both MNIST and CIFAR-10 in Figure 1. To interpret the result, for example on MNIST, when $\sigma = 0.4$, Algorithm 1 achieves at least $50\%$ accuracy under attacks whose $\ell_2$-norm sizes, $\|\mathbf{x} - \mathbf{x}'\|_2$, are smaller than $0.8$.

A clear trade-off between the tolerable attack sizes and the accuracy lower bound is present in Figure 1. This is anticipated, as our lower bound 6 indicates a higher standard deviation $\sigma$ results in a higher proportion of robust classification, but in practice, a larger amount of noise would also lead to a worse classification accuracy.

### 6.2   EMPIRICAL RESULTS

We next perform classification and measure the accuracy on real adversarial examples to evaluate the performance of our attack and defense methods. We first apply our attack method to the state-of-the-

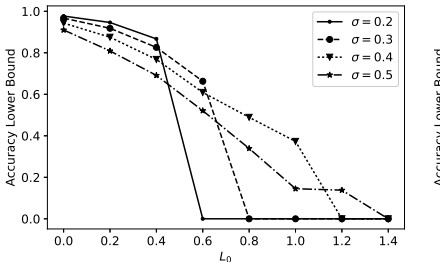 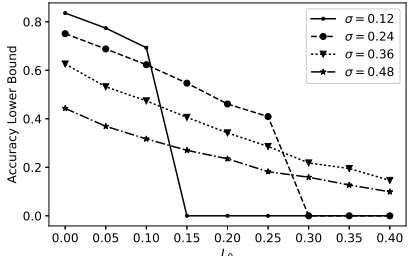

Figure 1: Accuracy lower bounds for **MNIST data set** (left) and **CIFAR-10 data set** (right) against tolerable sizes of attacks with various choices of $\sigma$ in Algorithm 1.

art defense model proposed in (Madry et al., 2017) based on adversarial training. In all experiments, we focus on five settings, summarized in Table 1. The adversarially trained model is trained against $\ell_\infty$ PGD attack while all the attacks are $\ell_2$ attack.

Table 1: Five settings of attack methods and defense models.

| Number | Defense Model | Attack Method |
|---|---|---|
| 1 | Naturally Trained Model | PGD |
| 2 | Adversarial Training (Madry's) | PGD |
| 3 | Naturally Trained Model | second-order (S-O) attack |
| 4 | Adversarial Training (Madry's) | second-order (S-O) attack |
| 5 | Stability trained model with Gaussian Noise (STN) | second-order (S-O) attack |

**MNIST** In the first plot in Figure 2, we monitor the average $\ell_2$ norm of gradients of the loss function during the construction of adversarial examples. Specifically, we compute $\frac{1}{N_{\text{batch}}} \sum_{i \in I} \| \nabla_{\mathbf{x}} L(\theta, \mathbf{x}, y) \big|_{\mathbf{x}_i^{(t)}} \|_2$ for each $t$, where $I$ is the index set of a batch. We monitor this quantity under the setting 1, 2 and 4 in Table 1. The result shows the $\ell_2$ norms of the gradients of the adversarially trained model are much smaller than the ones of the naturally trained model, validating our explanation in Section 3, that an adversarially trained model tends to make the loss function flat in the neighborhood of natural examples. It also shows our attack method can find adversarial examples with large loss more efficiently, by incorporating second-order derivative information.

In the second plot, we show the classification accuracy on MNIST under the setting 1, 2 and 4. The plot suggests that S-O attack is able to dramatically reduce the accuracy of Madry's model compared to PGD attack. The resulting accuracy is even worse than the one from the naturally trained baseline model. Note we do not include other attack methods such as C&W attack (Carlini & Wagner, 2017b) as no first-order attack is significantly stronger than PGD (Madry et al., 2017). We include the results of other attack methods in the Appendix.

In the third plot, we show the accuracy of different defense models under S-O attack, under the setting 3, 4 and 5. The plot suggests our model achieves better accuracy than both the baseline and the Madry's model.

Note in all experiments for evaluating our defense model, we use S-O attack as the adversary. The reason is if we use other attacks such as PGD, our defense model may cause the problem of obfuscating true gradients via randomization (Athalye et al., 2018), thus achieving robustness unfairly. On the other hand, as we discussed in Section 3, S-O attack is equivalent to EOT attack with respect to Gaussian noise (the noise we used in our defense model), thus it avoids obfuscating the true gradients via randomization. As a result, our results are reliable.

**CIFAR-10** In Figure 3, we show the classification accuracy on CIFAR-10 under the setting 3, 4 and 5. In addition, we include results for PixelDP from (Mathias et al., 2018) to show how stability training helps improve classification accuracy. Here the attack method we use is the S-O attack. As a comparison, our defense model obtains higher accuracies than both Madry's model and PixelDP against $\ell_2$ and $\ell_\infty$ attacks.

Overall, stability training combined with Gaussian noise shows a promising level of robustness and performs better than other models even under attacks that incorporate randomization.

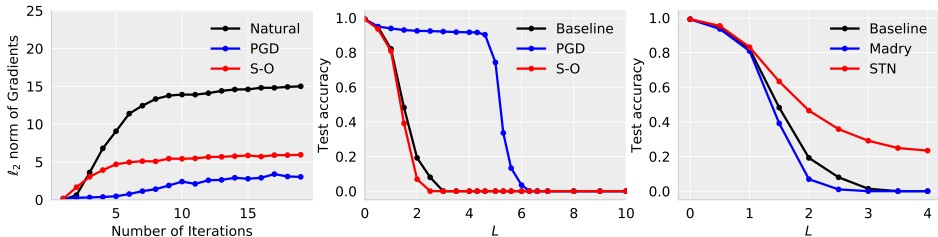

Figure 2: **MNIST data set** Left: Average $\ell_2$ norm of the gradients of the loss function for a batch in each iteration during adversarial attack. Middle: Classification accuracy for Madry's model under PGD attack and S-O attack. Right: Accuracy for different models under the same S-O attack.

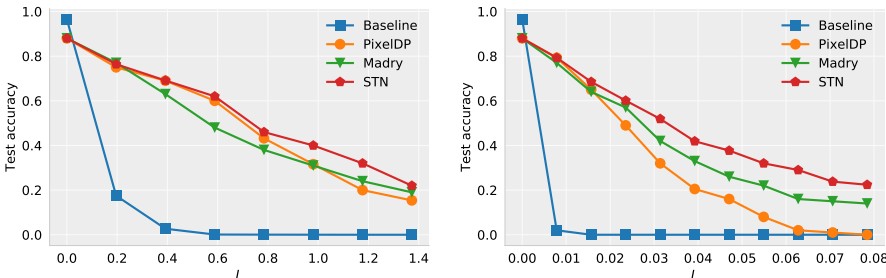

Figure 3: CIFAR-10 data set. We compare the accuracy on adversarial examples with various attack sizes for both $\ell_2$ (left) and $\ell_\infty$ (right).

Another strength of our method is that stability training only requires twice the computational time, whereas adversarial training is extremely time-consuming due to the iterative construction of adversarial examples.

## 7 DISCUSSION

**Adversarial Training Overfit to the Choice of Norms** In (Madry et al., 2017), the authors propose an adversarially trained model and argue it is robust against $\ell_2$ attacks even though it is trained against $\ell_\infty$ attacks. Our proposed $\ell_2$ attack significantly reduces their classification accuracy. On the other hand, the $\ell_\infty$ version of S-O attack also successfully attacks the adversarially trained model based on $\ell_2$ attacks. We include this experiment result in the Appendix. In general, we find adversarial training overfit to the choice of norms.

**On the Gap between Empirical and Theoretical Results** A noticeable gap exists between the theoretical bound shown in Figure 1 and the empirical accuracy. There are three possible explanations for this gap, each pointing to a direction for future works.

The first explanation for the gap is that the proposed upper bound is not tight. Another possible reason is that the empirical results should be worse for stronger attacks that have not been proposed, as it has happened to many defense models. The third one is due to the limitation of the $\ell_2$ distance. Although our framework is proposed in the context of adversarial attacks, where potential attacks are limited to be non-perceptible by humans, our theoretical analysis does not distinguish the types of perturbations up to their $\ell_2$ norms. In practice, one can perturb a few pixels with large values, such that the change is perceptible by humans and indeed leads to a change in classification, but it only yields small $\ell_2$ distance. The existence of such perturbations enforces the small upper bound. Future work might explore similar guarantees for the stronger $\ell_\infty$ distance.

## 8 CONCLUSION

We propose S-O attack, a new attack method based on an approximated second-order derivative of the loss function. S-O attack can effectively attack adversarial training, a training strategy that demonstrated significant robustness in previous works. We also propose an analysis on constructing defense models with certifiable robustness, combined with a strategy based on stability training for improving the robustness of the defense models. The experimental results show our defense model achieve better robustness compared to previous works.

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

## A  FAST APPROXIMATE METHOD MIYATO ET AL. (2017)

Power iteration method (Golub & Van der Vorst, 2001) allows one to compute the dominant eigenvector $\mathbf{r}$ of a matrix $\mathbf{H}$. Let $\mathbf{d}^0$ be a randomly sampled unit vector which is not perpendicular to $\mathbf{r}$, the iterative calculation of

$$\mathbf{d}^{t+1} = \frac{\mathbf{H}\mathbf{d}^t}{\|\mathbf{H}\mathbf{d}^t\|_2}$$

leads to $\mathbf{d}^t \to \mathbf{r}$. Given $\mathbf{H}$ is the Hessian matrix of $L(\theta, \mathbf{x}, y)$, we further use finite difference method to reduce the computational complexity:

$$\mathbf{H}\mathbf{d} \approx \frac{\nabla_{\mathbf{x}+\xi\mathbf{d}} L(\theta, \mathbf{x} + \xi\mathbf{d}, y) - \nabla_{\mathbf{x}} L(\theta, \mathbf{x}, y)}{\xi}$$
$$= \frac{\nabla_{\mathbf{x}+\xi\mathbf{d}} L(\theta, \mathbf{x} + \xi\mathbf{d}, y)}{\xi}$$

where $\xi > 0$ is the step size. If we only take one iteration, it gives an approximation that only requires the first-order derivative:

$$\mathbf{r} \approx \frac{\mathbf{H}\mathbf{d}}{\|\mathbf{H}\mathbf{d}^t\|_2} \approx \frac{\nabla_{\mathbf{x}+\xi\mathbf{d}} L(\theta, \mathbf{x} + \xi\mathbf{d}, y)}{\|\nabla_{\mathbf{x}+\xi\mathbf{d}} L(\theta, \mathbf{x} + \xi\mathbf{d}, y)\|}$$

which gives equation 3.

## B  PROOF OF LEMMA 1

**Lemma 1** Let $P = (p_1, \ldots, p_k)$ and $Q = (q_1, \ldots, q_k)$ be two multinomial distributions over the same index set $\{1, \ldots, k\}$. If the indexes of the largest probabilities do not match on $P$ and $Q$, that is $\mathrm{argmax}_i\, p_i \neq \mathrm{argmax}_j\, q_j$, then

$$D_\alpha(Q\|P) \geq -\log\left(1 - p_{(1)} - p_{(2)} + 2\left(\frac{1}{2}\left(p_{(1)}^{1-\alpha} + p_{(2)}^{1-\alpha}\right)\right)^{\frac{1}{1-\alpha}}\right) \tag{7}$$

where $p_{(1)}$ and $p_{(2)}$ are the largest and the second largest probabilities in $p_i$'s.

**Proof** Think of this problem as finding $Q$ that minimizes $D_\alpha(Q\|P)$ such that $\mathrm{argmax}\, p_i \neq \mathrm{argmax}\, q_i$ for fixed $P = (p_1, \ldots, p_k)$. Without loss of generality, assume $p_1 \geq p_2 \geq \cdots \geq p_k$.

It is equivalent to solving the following problem:

$$\min_{\sum q_i = 1, \mathrm{argmax}\, q_i \neq 1} \frac{1}{1-\alpha} \log\left(\sum_{i=1}^{k} p_i \left(\frac{q_i}{p_i}\right)^\alpha\right)$$

As the logrithm is a monotonically increasing function, we only focus on the quantity $s(Q\|P) = \sum_{i=1}^{k} p_i \left(\frac{q_i}{p_i}\right)^\alpha$ part for fixed $\alpha$.

We first show for the $Q$ that minimizes $s(Q\|P)$, it must have $q_1 = q_2 \geq q_3 \geq \cdots \geq q_k$. Note here we allow a tie, because we can always let $q_1 = q_1 - \epsilon$ and $q_2 = q_2 + \epsilon$ for some small $\epsilon$ to satisfy $\mathrm{argmax}\, q_i \neq 1$ while not changing the Renyi-divergence too much by the continuity of $s$.

If $q_j > q_i$ for some $j \geq i$, we can define $Q'$ by mutating $q_i$ and $q_j$, that is $Q' = (q_1, \ldots, q_{i-1}, q_j, q_{i+1} \ldots, q_{j-1}, q_i, q_{j+1}, \ldots, q_k)$, then

$$s(Q\|P) - s(Q'\|P) = p_i \left( \frac{q_i^\alpha - q_j^\alpha}{p_i^\alpha} \right) + p_j \left( \frac{q_j^\alpha - q_i^\alpha}{p_j^\alpha} \right) = (p_i^{1-\alpha} - p_j^{1-\alpha})(q_i^\alpha - q_j^\alpha) > 0$$

which conflicts with the assumption that $Q$ minimizes $s(Q\|P)$. Thus $q_i \geq q_j$ for $j \geq i$. Since $q_1$ cannot be the largest, we have $q_1 = q_2 \geq q_3 \geq \cdots \geq q_k$.

Then we are able to assume $Q = (q_0, q_0, q_3, \ldots, q_k)$, and the problem can be formulated as

$$\min_{q_0, q_2, \ldots, q_k} p_1 \left( \frac{q_0}{p_1} \right)^\alpha + p_2 \left( \frac{q_0}{p_2} \right)^\alpha + \sum_{i=3}^{k} p_i \left( \frac{q_i}{p_i} \right)^\alpha$$

$$\text{subject to} \quad 2q_0 + q_3 + \cdots + q_k = 1$$
$$\text{subject to} \quad q_i - q_0 \leq 0 \quad i \geq 1$$
$$\text{subject to} \quad -q_i \leq 0 \quad i \geq 0$$

which forms a set of KKT conditions. Using Lagrange multipliers, one can obtain the solution $q_0 = \left( \frac{p_1^{1-\alpha} + p_2^{1-\alpha}}{2} \right)^{\frac{1}{1-\alpha}}$ and $q_i = \frac{1-2q_0}{1-p_1-p_2} p_i$ for $i \geq 3$. Plug in these quantities, the minimized Renyi-divergence is

$$-\log \left( 1 - p_1 - p_2 + 2 \left( \frac{1}{2} \left( p_1^{1-\alpha} + p_2^{1-\alpha} \right) \right)^{\frac{1}{1-\alpha}} \right)$$

Thus, we obtain the lower bound of $D_\alpha(Q\|P)$ for $\text{argmax} p_i \neq \text{argmax} q_i$. ∎

## C  PROOF OF THEOREM 2

A simple result from imformation theory:

**Lemma 3** *Given two real-valued vectors $\mathbf{x}_1$ and $\mathbf{x}_2$, the Rényi divergence of $N(\mathbf{x}_1, \sigma^2 I)$ and $N(\mathbf{x}_2, \sigma^2 I)$ is*

$$D_\alpha(N(\mathbf{x}_1, \sigma^2 I)\|N(\mathbf{x}_2, \sigma^2 I)) = \frac{\alpha\|\mathbf{x}_1 - \mathbf{x}_2\|_2^2}{2\sigma^2} \tag{8}$$

**Theorem 2** Suppose we have $\mathbf{x} \in \mathcal{X}$, and a potential adversarial example $\mathbf{x}' \in \mathcal{X}$ such that $\|\mathbf{x} - \mathbf{x}'\|_2 \leq L$. Given a k-classifier $f : \mathcal{X} \to \{1, \ldots, k\}$, let $f(\mathbf{x} + N(\mathbf{0}, \sigma^2 I)) \sim (p_1, \ldots, p_k)$ and $f(\mathbf{x}' + N(\mathbf{0}, \sigma^2 I)) \sim (p'_1, \ldots, p'_k)$.

If the following condition is satisfied, with $p_{(1)}$ and $p_{(2)}$ being the first and second largest probabilities in $p_i$'s:

$$\sup_{\alpha>1} \left( -\frac{2\sigma^2}{\alpha} \log \left( 1 - 2M_1 \left( p_{(1)}, p_{(2)} \right) + 2M_{1-\alpha} \left( p_{(1)}, p_{(2)} \right) \right) \right) \geq L^2 \tag{9}$$

then $\text{argmax}_i \, p_i = \text{argmax}_j \, p'_j$

**Proof** From lemma 3, we know for $\mathbf{x}$ and $\mathbf{x}'$ such that $\|\mathbf{x} - \mathbf{x}'\|_2 \leq L$, with a k-class classification function $f : \mathcal{X} \to \{1, \ldots, k\}$:

$$D_\alpha(f(\mathbf{x}' + N(\mathbf{0}, \sigma^2))\|f(\mathbf{x} + N(\mathbf{0}, \sigma^2))) \leq D_\alpha(\mathbf{x}' + N(\mathbf{0}, \sigma^2)\|\mathbf{x} + N(\mathbf{0}, \sigma^2)) \leq \frac{\alpha L^2}{2\sigma^2} \tag{10}$$

if $N(\mathbf{0}, \sigma^2)$ is a standard Gaussian noise. The first inequality comes from the fact that $D_\alpha(Q\|P) \geq D_\alpha(g(Q)\|g(P))$ for any function $g$.

Therefore, if we have

$$-\log\left(1 - 2M_1\left(p_{(1)}, p_{(2)}\right) + 2M_{1-\alpha}\left(p_{(1)}, p_{(2)}\right)\right) \geq \frac{\alpha L^2}{2\sigma^2} \tag{11}$$

It implies

$$D_\alpha(f(\mathbf{x}' + N(\mathbf{0}, \sigma^2))\|f(\mathbf{x} + N(\mathbf{0}, \sigma^2))) \leq -\log\left(1 - 2M_1\left(p_{(1)}, p_{(2)}\right) + 2M_{1-\alpha}\left(p_{(1)}, p_{(2)}\right)\right) \tag{12}$$

Then from Lemma 1 we know that the index of the maximums of $f(\mathbf{x} + N(\mathbf{0}, \sigma^2))$ and $f(\mathbf{x}' + N(\mathbf{0}, \sigma^2))$ must be the same, which means they have the same prediction, thus implies robustness. ∎

## D  COMPARISON BETWEEN BOUNDS

We use simulation to show our proposed bound is higher than the one from PixelDP (Mathias et al., 2018).

In PixelDP, the upper bound for the size of attacks is indirectly defined: if $p_{(1)} \geq e^{2\epsilon}p_{(2)} + (1 + e^\epsilon)$ and the added noise has the distribution $N(0, \sigma^2 I)$, then the classifier is robust against attacks whose $\ell_2$ size is less than $\frac{\sigma\epsilon}{\sqrt{2\log(1.25/\delta)}}$.

As both bounds are determined by the models and data only through $p_{(1)}$ and $p_{(2)}$, it is sufficient to compare them with simulation for different $p_{(1)}$ and $p_{(2)}$ as long as $p_{(1)} \geq p_{(2)} \geq 0$, $p_{(1)} + p_{(2)} \leq 1$ and $p_{(1)} + p_{(2)} \geq 0.2$ are satisfied.

For fixed $\sigma$, $\epsilon$ and $\delta$ are two tuning parameters that affect the result. For a fair comparison, we use a grid search to find $\epsilon$ and $\delta$ that maximizes their bound.

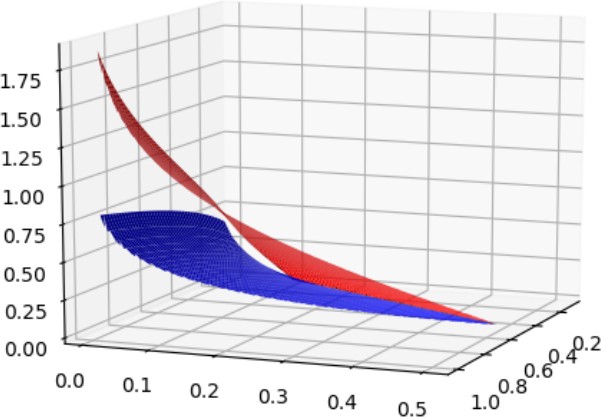

Figure 4: The upper bounds under different $p_{(1)}$ and $p_{(2)}$. In both settings, we let the variance of Gaussian noise $\sigma^2 = 1$. Our bound (red) is strictly higher than the one from PixedDP(blue).

The simulation result shows our bound is strictly higher than the one from PixelDP.

## E  COMPARISON TO OTHER ATTACK METHODS

We first compare our attack method to well-known C&W attack (Carlini & Wagner, 2017b) on MNIST in FIgure 5.

We then present experimental results from (Schott et al., 2018) for various existing attack methods with the size of attacks $\ell_2 = 1.5$, and compare them to our S-O attack in Table 2. All previous

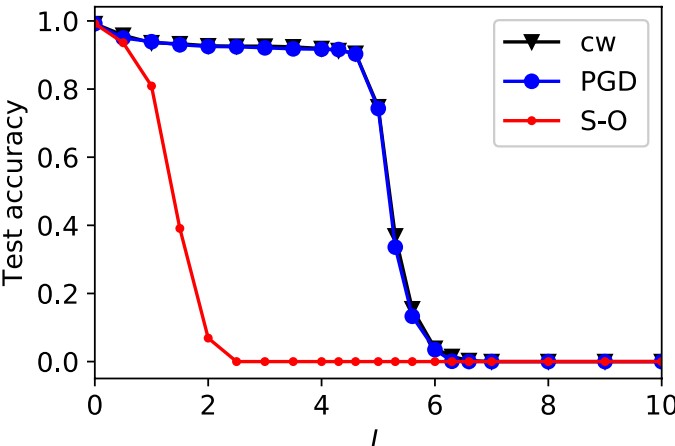

Figure 5: S-O attack compared to C&W attack on **MNIST** under various $\ell_2$ sizes.

methods, including FGSM (Kurakin et al., 2016), DeepFool (Moosavi-Dezfooli et al., 2016), and $\ell_2$ version PGD (Madry et al., 2017) cannot significantly reduce the accuracy of Madry's model.

The only attack method that achieves similar performance is Boundary Attack proposed by Schott et al. (2018), and we recognize it as concurrent work. Boundary attack iteratively flips pixels between the perturbed value and its original value to explore the minimum perturbation for altering the classification results. Although it produces a small number of perturbations, one major issue of the Boundary Attack is its computational inefficiency. It is hard to scale up to even **CIFAR10**.

Table 2: Different attack methods on Madry's model.

|  | FGSM | DeepFool | PGD | Boundary Attack | S-O (ours) |
|---|---|---|---|---|---|
| Madry's Model | 96% | 91% | 88% | 37.0% | 39.5% |

## F  $L_\infty$ S-O Attack Breaks Adversarial Training via $L_2$ Attacks

We apply $L_\infty$ S-O attack to Madry's models based on $L_\infty$ and $L_2$ attacks respectively. The accuracy is plotted in Figure 6. It is clear that only the model trained against $L_\infty$ attack is robust to the $L_\infty$ attack. This result supports our claim that adversarial training overfits to the choice of norm.

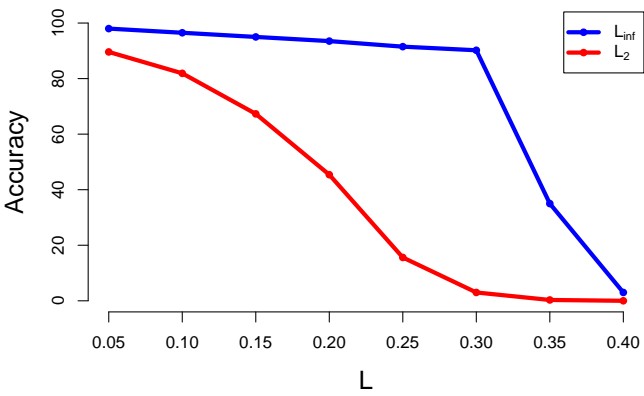

Figure 6: Accuracy of adversarial training via $L_\infty$ and $L_2$ attacks against $L_\infty$ S-O attack. The model based on $L_2$ is clearly vulnerable to $L_\infty$ attacks.

## G  ADVERSARIAL EXAMPLES ILLUSTRATION

We illustrate some randomly selected perturbed adversarial examples with size $\ell_2 = 2.0$ in figure 7. One can observe that there are a limited amount of perturbations in the adversarial examples, yet adversarially trained model gives false predictions for all examples.

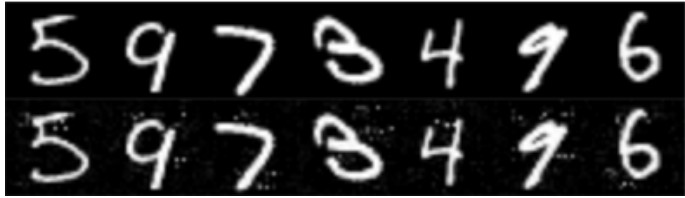

Figure 7: **Above:** Natural examples from MNIST. The correct labels are 5,9,7,3,4,9,6. **Below:** Adversarial examples with perturbation size $\ell_2 = 2.0$. The adversarially trained model predictions are 3,4,2,8,9,4,5.

## H  IMPLEMENTATION DETAILS

In this section, we specify the hyperparameters used in our experiments and other details. For all experiments, the baseline models are implemented using the codes from <https://github.com/MadryLab/mnist_challenge> and <https://github.com/MadryLab/cifar10_challenge>.

Our defense model only requires the following modification: 1) For stability training, we remove the adversarial training part and include the stability training regularizer. 2) At test time, we add i.i.d. Gaussian noise to pixels before feeding the images into the model.

For MNIST, we use stability training with STD of the Gaussian noise being $\sigma_{\text{train}} = 0.2$. The STD of Gaussian noise during testing is $\sigma_{\text{test}} = 0.2$. For CIFAR-10, the STD of Gaussian noise in stability training $\sigma_{\text{train}} = \frac{100}{255}$. The STD of Gaussian noise during testing is $\sigma_{\text{test}} = \frac{50}{255}$. Weight of the regularizor $\gamma = 1$ for both data sets.

