# OpenReview forum: "Second-Order Adversarial Attack and Certifiable Robustness"
_ICLR.cc/2019/Conference_

### Official Review · AnonReviewer1 · 2018-10-28
**The paper lacks clarity and needs to better contrast their work to existing results**

**Rating:** 3
**Confidence:** 5

**Review:**

This paper makes two different contributions in the field of adversarial training and robustness.
First the authors introduce a new type of attack that exploits second-order information while traditional attacks typically rely on first-order information.
Another contribution is a theorem that using the Renyi divergence certifies robustness of a classifier by adding Gaussian noise to pixels.

Overall, I find that the paper lacks clarity and does not properly contrast their work to existing results. They are also some issues with the evaluation results. I provide detailed feedback below.

1) Prior work
a) Connection between adversarial defense and robustness to random noise
This connection is established in Fawzi, A., Moosavi-Dezfooli, S. M., & Frossard, P. (2016). Robustness of classifiers: from adversarial to random noise. In Advances in Neural Information Processing Systems (pp. 1632-1640).
b) Connection between minimal perturbation required to confuse classifier and its confidence was discussed for the binary classification in Section 4 of
Fawzi, Alhussein, Omar Fawzi, and Pascal Frossard. "Analysis of classifiers’ robustness to adversarial perturbations." Machine Learning 107.3 (2018): 481-508.
c) The idea to compute the distribution of classifier outputs when the input is convolved with Gaussian noise was already “anticipated” in Section V of the following paper which relates the minimum perturbation needed to fool a model to it’s misclassification rate under Gaussian convolved input:
Lyu, Chunchuan, Kaizhu Huang, and Hai-Ning Liang. "A unified gradient regularization family for adversarial examples." Data Mining (ICDM), 2015 IEEE International Conference on. IEEE, 2015.

These papers should be discussed in the paper, please elaborate how you see your contribution regarding the results derived there.

2) Second-order attack introduced in the paper
I think they are a number of important details that are ignored in the presentation.
a) Regarding the assumption that the gradient vanishes in the difference of the loss, I think the authors should elaborate as to why this is a reasonable assumption to make. If we assume that the classifier has been trained to optimality then expanding the function at this (near-)optimum would perhaps indeed yield to a gradient term of small magnitude (assuming the function is smooth). However, nothing guarantees that the magnitude of the gradient term is negligible compared to the second-order information. The boundary of the classifier could very well be in a region of low-curvature.
b) The approximation of the second-order information is rather crude. However, the update derived is very similar to PGD with additional noise. In optimization, the use of noise is known to extract curvature, see e.g. (Xu & Yang, 2017) who showed that noisy gradient updates act as a noisy Power method that extracts negative curvature direction.
Xu, Y., & Yang, T. (2017). First-order Stochastic Algorithms for Escaping From Saddle Points in Almost Linear Time. arXiv preprint arXiv:1711.01944.

3) Issue of "degenerate global minimum": The authors argue that multistep attacks also suffer from this issue. However, the PGD attack of Madry is also initialized at a random point within the uncertainty ball around x, i.e. PGD attack first adds random noise to x before iteratively ascending the loss function. This PGD update + noise at first iteration seem rather similar to the update derived by the authors that uses random noise at every iteration. It could therefore be that the crude approximation of second-order information is not so different from previous work. This should be further investigated either theoretically or empirically.

4) Lack of details regarding some important aspects in the paper
a) “Note the evaluation requires adjustment and computing confidence intervals for p(1) and p(2), but we omit the details as it is a standard statistical procedure”
The authors seem to sweep this under the carpet but this estimation procedure gives only an estimate of the required quantities p(1) and p(2), which I think would require adjusting the result in the theorem to be a high probability bound (or an expectation bound) instead of a deterministic result.

b) “the noise is not necessarily added directly to the inputs but also to the first layer of a DNN. Given the Lipschitz constant of the first layer, one can still calculate an upper bound using our analysis. We omit the details here for simplicity”
What exactly changes here? How do you estimate the Lipschitz constant in practice?


5) Main theorem needs to be contrasted to previous results
The main Theorem uses the Renyi divergence certifies robustness of a classifier by adding Gaussian noise to pixels. There are already many results in the field of robust optimization that already derive similar results, see e.g.
Namkoong, H., & Duchi, J. C. (2017). Variance-based regularization with convex objectives. In Advances in Neural Information Processing Systems (pp. 2971-2980).
Gao, R., & Kleywegt, A. J. (2016). Distributionally robust stochastic optimization with Wasserstein distance. arXiv preprint arXiv:1604.02199.
Can you elaborate on the difference between your bounds and these ones? You do mention some of them require strong assumptions such as smoothness but this actually seems like a mild assumption (although some activation functions used in neural nets are indeed not smooth).

6) Adversarial Training Overfit to the Choice of norms
The main theorem derived in the paper uses the l_2 norm. What can be said regarding other norms?

7) Experiments:
a) the authors only report accuracies for attacks whose l2-norm is smaller than a fixed constant 0.8. However, this makes the results difficult to interpret and the authors should instead state the signal to noise ratio, i.e. dividing the l2-norm of the perturbation by the l2-norm of the image. Otherwise, it is not clear how strong or weak such perturbations are. (In particular, the norm depends on the dimension of the image, so l2-norms of perturbations for MNIST and CIFAR10 are not comparable).
b) In Section 6.2, the authors state that an l_infty trained model is vulnerable against l_2 perturbations. Why not training the model under both l_infty and l_2 perturbations?
c) Figure 1
Based on the results predicted in Theorem 2, it seems it would be more interesting to evaluate the largest L for which the classifier predictions are the same. Why did you report a different results?

8) Other comments
section 2.1: “Note this distribution is different from the one generated from softmax”. Why/How is this different?
connection to EOT attack’: authors claim: E_{d∼N(0,σ2I)} [∇_x L(θ, x, y)|x+d] = ∇_x E_{d∼N(0,σ2I)} [∇_x L(θ, x, y)|x+d]. There is a typo on the RHS where ∇_x is repeated twice. This is also the common reparametrization trick so could cite
Kingma, D. P., & Welling, M. (2013). Auto-encoding variational bayes. arXiv preprint arXiv:1312.6114.

---

> ### Author Response · Authors · 2018-11-06
> **Author's response 1**
>
> We thank the reviewer for the valuable comments. We respond to the questions and concerns in the following:
>
> 1)
>
> a) Our conclusion that improving robustness against to Gaussian random noise helps improve robustness against adversarial robustness is consistent with the results from Fawzi et al. (2016). In (13), they proved that the random robustness is upper bounded by the adversarial robustness under certain conditions. On the other hand, our conclusion comes from the derived bound (6) in our paper.
>
> b) In Fawzi et al. (2018), the theoretical results are derived under binary classification task with linear or quadratic classifiers. Our results are applicable to all kinds of classifiers. The intuition in this paper that more flexible classifiers achieve better robustness is partially consistent with our results. In our paper, we showed that a classifier with more confident classification (higher p_(1) and lower p_(2)) achieve better robustness.
>
> c) Adding some amount of Gaussian noise to reduce the effect of adversarial perturbations is intuitively rather straightforward. Their paper did not provide rigorous theoretical justification for their method but only reported empirical results. We proved why Gaussian noise can actually provide robustness that any adversarial attack cannot break.
>
>
> 2)
>
> a) The gradient vanishing is validated in Figure 1, where we show the magnitude of the gradients for adversarial examples is much smaller than natural examples. The effectiveness of our attack is an evidence that the second-order information is not negligible. Otherwise, utilizing such information should not provide any merit. Our experiments show that introducing second-order information not only increase the magnitude of the gradients in the following steps but also make our attack strong enough to break adversarial training.
>
> b) Using noise to extract curvature is exactly what we are doing here. Please see the proof in Appendix A. We will cite this paper as related work.
>
> 3) One major difference between our method and PGD with random noise is that at each step, we add noise multiple times and update the example with the average gradient. If we only add noise once as a noisy PGD, it is equivalent to estimating the expectation of a distribution with only one sample, which is not sufficient. It is essential to add noise multiple times and take the average to precisely estimate the approximated second order information. Empirically, a noisy PGD cannot break adversarial training. We will add an experiment to show this.
>
> 4) We do not include these two details as they were fully discussed in Lecuyer, Mathias, et al. (2018). The first point is just estimating the confidence intervals of a multinomial distribution with i.i.d. samples, which is a standard statistical procedure. We will state it more explicitly in the algorithm in the revised version. For the second point, if we know the Lipschitz constant, one can think of the outputs of the first layer as the inputs and redo the analysis. The divergence between the new inputs is bounded by the Lipschitz constant multiplied by the divergence between the original inputs, as the divergence is just the L_2 norm. The estimation of the Lipschitz constant differs for different models, and we refer to Lecuyer, Mathias, et al. (2018) for details, as this is not the major focus of our paper.
>
> Lecuyer, Mathias, et al. "On the Connection between Differential Privacy and Adversarial Robustness in Machine Learning." arXiv preprint arXiv:1802.03471 (2018).

---

> ### Author Response · Authors · 2018-11-06
> **Author response 2**
>
> 5) There are indeed many results for robust optimization and certifiable robustness. The main advantages of our method are a) Gao et al. (2016) derives a more general result for optimization over a Wasserstein ball of a nominal distribution. Namkoong et al. also study the robust optimization over a \phi-divergence ball of a nominal distribution. In the literature, the divergence between distribution is barely used to empirically evaluate the strength of adversarial attacks. On the other hand, our analysis utilizes the fact that the Renyi divergence between two equal variances Gaussian distribution is a function of the L_2 norm of the difference of their means to derive a bound that can directly be applied to adversarial attack problem. b) It requires nothing on the structure of the classifiers. c) It is easy to compute. One can obtain the bound by multiple feedforward computations. Wong et al. (2017), for example, requires training an additional network.
>
> Kolter, J. Zico, and Eric Wong. "Provable defenses against adversarial examples via the convex outer adversarial polytope." arXiv preprint arXiv:1711.00851 2.4 (2017).
>
> 6) Yes. If we use Laplacian noise instead of Gaussian noise, one can derive similar results for the L_1 norm, as the Renyi divergence of the Laplace random variables can be expressed as a function of the L_1 difference of their means. Unfortunately, there is no distribution that leads to a bound for L_inf norm.
>
> 7)
>
> a) The use of the L_2 norm is quite standard in the literature, for example, see Madry et al. (2017). To see how strong or weak the perturbations are, we showed perturbed examples in Appendix G. We will add more illustrations for CIFAR10. We will also report the signal to noise ratio as we agree it is a reasonable concern.
>
> Madry, Aleksander, et al. "Towards deep learning models resistant to adversarial attacks." arXiv preprint arXiv:1706.06083 (2017).
>
> b) We thank the reviewer's great suggestion. We will add results with mixed adversarial examples added during training in the revised version.
>
> c) It is because each input has its own L as they have different p_(1) and p_(2). Therefore, it is difficult to illustrate L for a data set. Instead, we set a threshold L_0 ahead and find how many examples have L that surpass this threshold to quantify the robustness with respect to a data set.
>
> 8) This distribution is generated by counting how many times our classifier gives a certain result when we run the feedforward procedure multiple times. Note each time it may give different results as we added noise at the beginning. After this, we can form a histogram, from which we calculate p_(1) and p_(2). The softmax is just the quantity we use to determine the class at each run. Please see step 1-5 in Algorithm 1.
>
> This is also related to point 4 where we mention estimating the confidence interval for p_(1) and p_(2) is just estimating the confidence interval for a multinomial distribution. From the distribution generating procedure above, it is clear that p_j's obey a multinomial distribution.
>
> We will fix the typo and cite relevant papers in the revised version.

---

### Official Review · AnonReviewer3 · 2018-11-02

**Rating:** 5
**Confidence:** 3

**Review:**

This paper consists of two parts: a 2nd-order attack method and a certification for robustness. The paper is well written and easy to follow. However, addressing of similarity and comparison with some previous methods could be improved.

First of all, the motivation of 2nd order attack is clear and reasonable: for adversarially trained model at minimax, the gradient is close to vanishing, so 2nd order information helps a lot to find actual adversarial examples in this case. However,

1. A lot of defenses have tried to modify the networks to make even computing the gradient difficult if not impossible. In this case, how effective is the 2nd order attack? I would like to see some discussion of this.

2. While the starting point seems like a powerful attack method. The 2nd order information is only approximately computed via finite differences. A powerful method with weak approximation will sounds more powerful than a weak method to start with, but the actual effectiveness will need more systematic comparison. I think adding some studies of the accuracy or variances of the 2nd order information with the proposed approximation method (under natural setting and maybe also under the setting where the networks are modified to make even 1st order information hard to compute) would definitely help.

3. Also after the approximation, as mentioned in the paper, the algorithm becomes equivalent to EOT attacks with Gaussian noises. The EOT attacks are also more general to allow different types of noises. While it might not make lots of sense to compare with EOT attacks in the experiments as the two algorithms seem to be exactly the same, it would help of more discussions could be devoted to justify how the proposed algorithm is novel given the previously existed EOT attack.

4. In the experiments on adversarially trained models, the adversarial trained models are trained against l_inf attack, while the actual attack is l2. This seems unfair. Since the author mentioned that it is easy to extend their method to l_inf attack. It would be more justifiable if the results with matching attack types are shown instead of the current ones.

The certified robustness is an interesting take, too. However, the bounds might be too strong: as far as I understand, it does not rely much on the properties of the underlying neural networks f. So in order to be applicable to all kinds of weird non-robust neural networks uniformly, the bounds cannot be too tight. To get useful certificate level, a too heavy noise level sigma might be needed and potentially destroys the classification accuracy of the original model f. This is acknowledged in the 'gap between theory and empirical' section. And it makes the importance of such kind of bounds a bit weak.

5. Also, the 'stability training' procedures derived based on this bounds is quite similar to some previous methods. For example, the objective function is very similar to 'logit pairing', which add an extra term to bound the similarity between two logits from an adversarial or noisy version. The empirical results will be much stronger if more closely related methods are included in the comparison. For example, logit pairing, as well as simple training with Gaussian perturbation on inputs.

In summary, this paper provide some interesting perspectives to adversarial attacks and certifications. However, the main algorithms are very similar to some existing methods, more discussion could be used to compare with the existing literature and clarify the novelty of the current paper. The empirical results could also be made more stronger by including more relevant baseline methods and more systematic study of the effectiveness of some approximation methods adopted

---

> ### Author Response · Authors · 2018-11-06
> **Author's response**
>
> We thank the reviewer for the valuable comments. We respond to the questions and concerns in the following:
>
> 1) The defense methods that try to make computing gradient difficult is called gradient obfuscating. This kind of defense has been shown vulnerable to stronger attacks. Please see https://arxiv.org/abs/1802.00420. Therefore, we do not discuss this kind of defense in our paper.
>
> 2) We believe the effectiveness of the second order information has been shown in Figure2, where we show using second order information increase the magnitude of the gradients and reduce the accuracy of Madry's model. The point of the second order attack is not to estimate the second order information accurately, but to utilize the approximated one to improve the effectiveness of our attack.
>
> 3) Although there is an equivalence between EOT and our attack, the motivation is totally different. EOT was proposed to attack defense models where randomness is present. The goal of EOT is to reduce the effect of randomness in a defense model by introducing the same randomness in the attack. In our attack, the randomness is used to reduce the effect of vanishing gradients and escape from "degenerate global minimum".
>
> 4) The point of our paper is to demonstrate that the robustness of adversarial training cannot generalize to different choices of norms, that is, if the model is adversarially trained against L_inf norm, then it is vulnerable to L_2 attack. In the paper, we called adversarial training was not *universally robust*. In practice, it is problematic because one does not know what kind of attacks will be used by the adversaries. Ideally, a robust model should be robust to all forms of attacks.
>
> We believe that the fact our bound does not rely on the property of the neural network is a strength rather than a weakness. There are works that show certifiable bounds when assuming the model is simply feedforward or Lipschitz-smooth or using some specific activation function, but cannot be extended to other models. Our analysis, on the other hand, is applicable to all different models, such as CNN, RNN, and even models that are not neural networks.
>
> In addition, to show the strength of our bound, as an example, a very recent paper https://arxiv.org/pdf/1811.00866v1.pdf (accepted by NIPS 2018) also proposes a certifiable bound of robustness (we did not cite this paper as it was released very recently). In Table 4 in their paper, one can see their L_2 norm bound is close to ours in Figure 1.
>
> 5) This form of object function has been studied by many papers. Our use of this form of the objective function is motivated by Zhang et al. (2016) Improving the Robustness of Deep Neural Networks via Stability Training. The goal of introducing this objective function is to improve the robustness against Gaussian noise, while "Logit Pairing" aims to improve the robustness against adversarial attacks. Logit Pairing has been shown vulnerable to adversarial attacks in https://arxiv.org/abs/1807.10272, therefore, we do not include this method. In addition, our method is equivalent to "simple training with Gaussian perturbation" when \gamma=0. We did not include the results for different choice \gamma, but we would like to add these results in the revised version.

---

### Official Review · AnonReviewer2 · 2018-11-05
**Interesting ideas, insufficient experimental evaluation.**

**Rating:** 4
**Confidence:** 5

**Review:**

The paper makes three rather independent contributions: a) a method for constructing adversarial examples (AE) utilizing second-order information, b) a method for certifying classifier robustness, c) a method to improve classifier robustness. I will discuss these three contributions separately.

a) Second order attack: Miyato et al. (2017) propose a method for constructing AE for the case where the gradient of the loss is vanishing. In this case, at given a point, the direction of steepest loss ascent can be approximated by the gradient at a randomly sampled nearby point. Miyato et al. (2017) show how this can be derived as a very crude approximation of the power method. The authors of the current paper apply this attack to the adversarial trained networks of Madry et al. (2017). They find that the *L_infinity* trained networks of that work are not as *L_2* robust as originally claimed. I find this result interesting, highlighting a failure case of first-order methods (PGD) for evaluating adversarial robustness. However, it is important to note that these were models that were *not* trained against an L2 attack and thus should not be expected to be very robust to one. Therefore, this result does not identify a failure of adversarial training as the authors seem to suggest but rather a failure of the original evaluation of Madry et al. (2017). It is also worth noting that this finding is specific to MNIST given the results currently presented. This might be explained by the fact that robust MNIST models tend to learn thresholding filters (Madry et al., 2017) which might cause gradient obfuscation.

b) Adversarial robustness certification: The authors proposed a method for certifying the robustness of a model based on the Renyi divergence. The core idea is to define a stochastic classifier that randomly perturbs the input before classifying it. Given such a classifier, one can construct the probability distribution over classes. The authors prove that given the gap between the first and second most likely classes, one can construct a bound on the L2 norm of perturbations required to fool the classifier. This method is able to certify the adversarial accuracy of some classifier to relatively small epsilon values. While I think the theoretical arguments are elegant, I find the overall contribution incremental given the work of Mathias et al. (2018). Both methods seem to certify robustness of roughly the same scale. One component of the experimental evaluation missing is how does the certifiable accuracy differ between robust and non-robust models. Currently there are only results for a single model (Figure 1) and it is not clear from the text which one it is. Given that there exists a section titled "improved certifiable robustness" I would at least expect a result where a model with higher certifiable accuracy is constructed.

c) Improved robustness via stability training: The authors propose a method to make a classifier more robust to input noise. They add a regulatization term to the training loss that penalizes a change in the probabilities predicted by the network when the input is randomly perturbed. In particular, they use the cross-entropy loss between the probability distributions predicted at the original and the perturbed point. The goal is to train a model that is more robust to random perturbation which will then hopefully translate to robustness to adversarial perturbation. This method is evaluated against the proposed attack (a) and is found to be more robust to that attack than previous adversarially trained models. Overall, I find the idea of stability training interesting. However I find the current evaluation severely lacking. First of all, these models should be evaluated against a standard PGD adversary (missing from Table 1). Even if that method is unreliable when applying random noise to the input at each step it is still an important sanity check. Additionally, in order to deal with the stochasticity of the model one should experiment with a PGD attack that estimates the gradient using multiple independent noise samples (see https://arxiv.org/abs/1802.00420). Finally, other attacks such as black-box attacks and finite-differences attacks should potentially be considered. Given how other defenses based purely on data augmentation during training or testing were bypassed it is important to apply a certain amount of care when evaluating the robustness of a model.

Overall, while I think the paper contains interesting ideas, I find the current evaluation lacking. I recommend rejection for now but I would be willing to update by score based on author responses.

Minor comments to the authors:
-- Last paragraph of first page: "Though successful in adversarial defensing, the underlying mechanism is still unclear.", adversarial training has a fairly principled and established underlying mechanism, robust optimization.
-- Figure 2 left: is the natural line PGD or SO?
-- The standard deviation of the noise used is very large relative to the pixel range. You might want to comment on that in the main text.
-- Figure 3: How was the Madry model trained? L_inf or L_2?

---

> ### Author Response · Authors · 2018-11-06
> **Author's response**
>
> We thank the reviewer for the valuable comments. We respond to the questions and concerns in the following:
>
> 1) We are not suggesting that there is a failure of adversarial training but trying to demonstrate that such training strategy cannot generalize in terms of the choice of norms. In the paper, we called adversarial training was not *universally robust*. In practice, it is problematic because one does not know what kind of attacks will be used by the adversaries. Ideally, a robust model should be robust to all forms of attacks. Our finding shows the importance of evaluating defense model under different norms, as many defense models only focus on L_inf norm.
>
> We thank the reviewer for pointing out that this phenomenon might be explained by the thresholding filters. Our second order attack used the information of the gradients, yet still successfully attacked the adversarial training model. If adversarial training on MNIST caused gradient obfuscation, any attack using gradient information cannot break it. Therefore, our result suggests adversarial training do not cause gradient obfuscation on MNIST.
>
> 2) The bound proposed in our paper is almost twice compared to the one from Mathias et al. (2018) as suggested in Figure 4. It is difficult to improve the robustness into a different scale. As an example, a very recent paper https://arxiv.org/pdf/1811.00866v1.pdf (accepted by NIPS 2018) also proposes a certifiable bound of robustness (we did not cite this paper as it was released very recently). In Table 4 in their paper, one can see the l_2 norm bound is in the same scale as ours.
>
> Note without stability training, our method is the same as the one (PixelDP) proposed by Mathias et al. (2018) *in practice*, that is adding Gaussian noise and pick the output with the highest probability. Therefore, the difference between PixelDP and STN is exactly the gain from stability training from the "improved certifiable robustness" section.
>
> 3) We would like to emphasize that the main points of our paper are 1) propose a new attack method that shows the weakness of adversarial training. 2) propose a certifiable defense framework that allows us to calculate a bound of robustness for a model.
>
> The empirical performance of our defense is not the major concern in this paper, although its performance seems to be better than other methods in our experiments. As we pointed out in the discussion, there is a large gap between the theoretical bound and the empirical results, and it is possible that this is caused by the overestimation of the empirical performance. In general, no matter how many experiments are performed, a defense without theoretical justification is still vulnerable to unknown attacks. This is actually the motivation of our certifiable defense framework.
>
> We understand that empirical evaluation is important as well. We will add more experimental results in the Appendix to help to understand the empirical performance of our method. However, as mentioned, the point of our defense is to show a certifiable bound of the robustness. In fact, many papers on certifiable defense only perform experiments on the theoretical bounds but not accuracies against attacks:
> https://arxiv.org/pdf/1710.10571.pdf
> https://arxiv.org/pdf/1811.00866v1.pdf
>
> minor: 1) We will change this sentence. What we meant is that there are some perspectives of adversarial training that are unclear.
> 2) it is the magnitude of the gradients of natural samples. The other two are the magnitude of the gradients of adversarial examples generated by PGD and SO respectively.
> 3) We will comment on that. Essentially a large noise will greatly hurt the classification accuracy, so it is important to keep a good balance.
> 4) It was trained in L_inf.

---

> > ### Comment · AnonReviewer2 · 2018-11-12
> > **Response**
> >
> > I appreciate the authors taking the time respond to my comments.
> >
> > 1) I understand the point about L_inf adversarial training not resulting in L_2 robustness. However, the way the paper is currently written, the emphasis is placed on SO attacks being able to bypass adversarial training by being more powerful than FO attacks. The difference between the norms used is only mentioned in the intro and the discussion and not at all in the main sections describing your contributions. I would suggest that this nuisance is emphasized in future versions of the paper.
> >
> > Note that the proposed SO attacks are not using gradient information at the current iterate, but are rather using gradient information from a random nearby point. This would explain why they might evade the flat regions caused by threshold filters on MNIST. Given that this is the only case where the proposed SO attacks is more powerful than standard PGD, I still believe that it is an artifact of the particular dataset and norm choice.
> >
> > 2) I understand that achieving certifiable robustness to a larger scale of perturbations is challenging. However, I personally view the results of Mathias et al. (2018) as "One can certify robustness to small epsilon values by adding random noise to the input". I view the result of the current paper as "One can do use a different toolkit to certify slightly larger epsilon values when adding random noise to the input". Hence while the approach and analysis are interesting, I find the result incremental.
> >
> > It is still not clear to me what the importance of stability training is. It would be helpful if the following comparisons were included: a) the certified bound using the approach of Mathias et al. (2018) _both_ for standard and stability-trained networks, b) the certified bound using the proposed approach _both_ for standard and STN. (The two approaches should be compared using the best setting of hyper-parameters for each.)
> >
> > 3) Even if the proposed defense is not one of the main contributions of the paper, it is still claimed as a contribution. There are several points in the paper where it is argued that the proposed defense outperforms other state-of-the-art methods and that "... our results are reliable.". If the evaluation happens to indeed be unreliable, then the claim could be misleading. This would be a significant flaw for a paper that cannot be ignored during the review process. If the empirical performance is indeed not a major concern, that it should be removed from the paper.

---

> > > ### Author Response · Authors · 2018-11-12
> > > **Response**
> > >
> > > We thank the reviewer for the thoughtful responses.
> > >
> > > 1) We will emphasize the point that our method broke adversarial learning only when different norms are used in the training and testing in the main sections.
> > >
> > > We agree that gradient obfuscation might exist from this perspective. This suggests that adversarial training does not truly achieve adversarial robustness on MNIST, which could be an important implication from our results.
> > >
> > > 2) It is true that we are doing the same task with a different tool, but the increment of the bound is significant instead of ''slightly''. From Figure 4 one can observe that we improve the bound from 0.75 to 1.75 when p_(1)->1 and p_(2)->0. In addition,t he tool we used is totally different from the one used in Matthias et al., and the proof is non-trivial.
> > >
> > > We want to clarify that our proposed method is exactly the approach of Mathias et al. (2018) combined with stability trained networks in practice, therefore the comparisons a) and b) are the same. We will add this comparison in the revised version to better illustrate the improvement.
> > >
> > > 3) We are not suggesting that the evaluation is not reliable. In the last rebuttal what we want to suggest is that empirical evaluation overall is not reliable in the sense that empirical evaluation does not lead to any formal proof for the observed robustness. For example, although adversarial training is widely acknowledged as the most successful defense method in the literature, there are already many new attacks (including ours) that reduce its robustness reported in the original paper.
> > >
> > > We will add more experiment results for different attacks to validate the robustness of our method.

---

### Meta-Review · Area_Chair1 · 2018-12-17
**Reject**

**Confidence:** 5
**Recommendation:** Reject

**Metareview:**

The reviewers have agreed this work is not ready for publication at ICLR.